# Characterization of Tumor-Infiltrating Lymphocyte-Derived Atypical TCRs Recognizing Breast Cancer in an MR1-Dependent Manner

**DOI:** 10.3390/cells13201711

**Published:** 2024-10-16

**Authors:** Abdul Hayee, Eiji Kobayashi, Chihiro Motozono, Hiroshi Hamana, Ha Thi Viet My, Takuya Okada, Naoki Toyooka, Satoshi Yamaguchi, Tatsuhiko Ozawa, Hiroyuki Kishi

**Affiliations:** 1Department of Immunology, Faculty of Medicine, Academic Assembly, University of Toyama, Toyama 930-0194, Japan; drahayee@med.u-toyama.ac.jp (A.H.); hiroshi.hamana@shinobitx.com (H.H.); drvietmy159@gmail.com (H.T.V.M.); satyamag@med.u-toyama.ac.jp (S.Y.); toza@eng.u-toyama.ac.jp (T.O.); 2Center for Advanced Antibody Drug Development, University of Toyama, Toyama 930-0194, Japan; 3Division of Infection and Immunity, Joint Research Center for Human Retrovirus Infection, Kumamoto University, Kumamoto 860-0811, Japan; motozono@kumamoto-u.ac.jp; 4Shinobi Therapeutics Co., Ltd., Kyoto 606-8304, Japan; 5Department of Biofunctional Molecular Chemistry, Faculty of Engineering, University of Toyama, Toyama 930-8555, Japan; tokada@eng.u-toyama.ac.jp (T.O.); toyooka@eng.u-toyama.ac.jp (N.T.); 6Department of First Internal Medicine, Faculty of Medicine, Academic Assembly, University of Toyama, Toyama 930-0194, Japan

**Keywords:** breast cancer, tumor-infiltrating lymphocytes, MR1, T-cell receptors

## Abstract

The MHC class I-related 1 (MR1) molecule is a non-polymorphic antigen-presenting molecule that presents several metabolites to MR1-restricted T cells, including mucosal-associated invariant T (MAIT) cells. MR1 ligands bind to MR1 molecules by forming a Schiff base with the K43 residue of MR1, which induces the folding of MR1 and its reach to the cell surface. An antagonistic MR1 ligand, Ac-6-FP, and the K43A mutation of MR1 are known to inhibit the responses of MR1-restricted T cells. In this study, we analyzed MR1-restricted TCRs obtained from tumor-infiltrating lymphocytes (TILs) from breast cancer patients. They responded to two breast cancer cell lines independently from microbial infection and did not respond to other cancer cell lines or normal breast cells. Interestingly, the reactivity of these TCRs was not inhibited by Ac-6-FP, while it was attenuated by the K43A mutation of MR1. Our findings suggest the existence of a novel class of MR1-restricted TCRs whose antigen is expressed in some breast cancer cells and binds to MR1 depending on the K43 residue of MR1 but without being influenced by Ac-6-FP. This work provides new insight into the physiological roles of MR1 and MR1-restricted T cells.

## 1. Introduction

The major histocompatibility complex (MHC) class I-like related molecule 1 (MR1) is a nonpolymorphic antigen-presenting molecule that associates with β2-microglobulin and presents antigenic molecules to unconventional T cells whose antigen recognition is not restricted to MHC class I molecules [1,2]. Although MR1 mRNA is widely expressed in various tissues and cells [3,4], MR1 protein expression is hardly detected on the cell surface [5]. McWilliam et al. found that for MR1 to fold and reach the cell surface, the binding of its ligand, such as acetyl-6-formyl pterin (Ac-6-FP), to MR1 molecules is required [6]. They also found that the formation of a Schiff base between the MR1 ligand and the K43 residue of MR1 is a prerequisite for MR1 folding and its exit from endoplasmic reticulum (ER).

Typical MR1-restricted T cells are mucosal-associated invariant T (MAIT) cells that react to riboflavin-synthesizing microbes in an MR1-restricted manner. Their T-cell receptor (TCR)Vα repertoire is almost invariant (TRAV1-2-TRAJ33), with a limited number of TCRVβ (TRBV20 and TRBV6) [7]. Identified antigenic molecules that activate MAIT cells include uracils (5-OP-RU) and lumazines. In contrast, pterins (Ac-6-FP, an acetylated derivative of the folic acid derivative 6-formylpterin (6-FP)) bind to MR1 molecules but do not activate MAIT cells [8,9]. In addition to typical MAIT cells, TRAV1-2^−^ MR1-restricted T cells that were detected by the 5-OP-RU/MR1 tetramer were reported [10]. Furthermore, self-reactive or cancer-reactive MR1-restricted T cells have been found in the blood of healthy individuals, whose reactivity was inhibited by 6-FP- or MR1-specific antibodies as MAIT cells [11,12,13,14]. However, their roles have not yet been elucidated.

Previously, we developed a method to rapidly and comprehensively analyze the TCR reactivity of tumor-infiltrating lymphocytes (TILs) to identify tumor cell-reactive TCRs without antigen information [15]. We designated this approach as comprehensive functional investigation of TCRs (c-FIT). Using c-FIT, we identified 15 TCRs that reacted to the MCF7 breast cancer cell line in an MR1-restricted manner from CD8^+^ TILs. Our MR1-restricted TCR reacted to some breast cancer cells but not to normal breast cells or other cancer cells. Furthermore, the reactivity of our TCR to breast cancer cells was not inhibited by Ac-6-FP- or MR1-specific antibodies. Our findings reveal that cancer-reactive MR1-restricted T cells infiltrate tumors in an antigen-dependent manner.

## 2. Materials and Methods

### 2.1. Reagents

Acetyl-6-FP was purchased from Cayman Chemicals (Ann Arbor, MI, USA) and dissolved in 10 mM NaOH at 2 mg mL^−1^. 5-Amino-6-(D-ribitylamino) uracil (5-A-RU) was purchased from Toronto Research Chemicals (North York, ON, Canada) and stored at –80 °C in solid form until it was dissolved in sterile H_2_O and frozen at −20 °C as a 10 mM stock solution. The 10 mM stock solution was combined with an equal volume of 50 μM methylglyoxal (diluted in sterile H_2_O) to produce 5-OP-RU. Peridinin chlorophyll protein-cyanin 5.5 (PerCP-Cy5.5)-conjugated anti-human CD3 monoclonal antibody (mAb) (45-0037-41), allophycocyanin (APC)-conjugated anti-human CD8 mAb (17-0088-41), APC-cyanin 7 (APC-Cy7)-conjugated Fixable Viability Dye (65-0865-14), and APC-conjugated anti-human CD8α mAb (17-0088-42) were purchased from eBioscience (Waltham, MA, USA). APC-conjugated anti-human HLA-A, B, C mAb (311410), APC-conjugated anti-human HLA-E mAb (342605), APC-conjugated anti-human CD80 mAb (305219), APC-conjugated (361108) and unconjugated (361102) anti-human/mouse/rat MR1 mAb (clone 26.5), APC-conjugated anti-human CD1a (300110), CD1b (329109), CD1c (331523), CD1d (350307), and APC-Cy7-conjugated anti-mouse CD3 (145-2C11) mAbs were purchased from BioLegend (San Diego, CA, USA). Human MR1 tetramers were purchased from MBL (Tokyo, Japan), and ligands, such as 5-OP-RU and Ac-6-FP, were loaded as per the instructions from the manufacturer. The pMXs-IRES-GFP retroviral expression vector (RTV-013) was acquired from Cell Biolabs (San Diego, CA, USA).

### 2.2. Cell Lines

BW5147.3 cells expressing mouse CD8αβ and CD3 were generously provided by Professor Ellis L. Reinherz, Dana Farber Cancer Institute, Harvard Institutes of Medicine, and maintained in Dulbecco’s modified Eagle’s medium (DMEM) containing 10% fetal calf serum (FCS), 2-mercaptoethanol (50 μM), streptomycin (100 μg mL^−1^), and penicillin (100 U mL^−1^) (DMEM culture medium) in the presence of 0.4 mg mL^−1^ hygromycin B and 0.4 mg mL^−1^ G418. Human CD8α and β cDNAs were transduced into BW5147.3 cells using retroviral vectors (named BW-hCD8αβ^+^ cells). MCF7 cells were purchased from ATCC (Manassas, VA, USA). The MCF7 cells were maintained in DMEM culture medium. Human CD80 and β2-microglobulin cDNAs were transduced into the MCF7 cells using retroviral vectors (human CD80^+^ β2-microglobulin^+^ MCF7 cells). Patient HLA-A, B, or C cDNA, together with green fluorescent protein (GFP) cDNA, was transduced into MCF7 cells using retroviral vectors, and GFP-positive cells were sorted as patient HLA-transduced cells using a FACSAria II (Becton Dickinson, Franklin Lakes, NJ, USA). To prepare luciferase-expressing MCF7 (MCF7-Luc), cells were prepared as described previously [15]. Briefly, luciferase cDNA (Promega, Madison, WI, USA), together with Kusabira-Orange cDNA (MBL), was transduced into the MCF7 cells using retroviral vectors. To select the MCF7-Luc cells, the Kusabira-Orange-positive cells were sorted using a FACSAria Ⅱ. The HLA class I (A, B, and C), β2-microglobulin, and MR1-deleted MCF7 cells were established as described previously [15]. Briefly, these cells were prepared by using the CRISPR-Cas9 KO plasmid according to the manufacturer’s instructions (Santa Cruz Biotechnology, Dallas, TX, USA). HLA class I and β2-microglobulin knockout cells were enriched by staining the cells with an anti-HLA-A, B, C mAb (W6/32), and the HLA class Ⅰ-negative cells were sorted using a FACSAria Ⅱ (MCF7ΔHLA-I, and MCF7Δβ2m). MR1 knockout cells were pulsed with Ac-6-FP and stained with anti-MR1 mAb. The MR1-negative cells were sorted (MCF7ΔMR1). MR1 deletion was confirmed by RT-PCR as described in the following section (RNA extraction and RT-PCR). K562, Jurkat, Karpas 299, COLO205, and A549 cell lines from laboratory cell stocks were used. A549, MDA-MB-231, MDA-MB-453, MDA-MB468, and BT474 cells were kindly provided Professor Yoshihiro Hayakawa, University of Toyama, and maintained in DMEM culture medium. The K562, Jurkat, Karpas 299, COLO205, and ZR-75-1 cells were maintained in RPMI 1640 containing 10% fetal calf serum (FCS), 50 μM 2-mercaptoethanol, streptomycin (100 μg mL^−1^), and penicillin (100 U mL^−1^) (RPMI culture medium). Phoenix-A cells and Plat-E cells were generously supplied by Professor Garry Nolan at Stanford University and Professor Toshio Kitamura, University of Tokyo, respectively, and maintained in DMEM culture medium. Human mammary epithelial cells (HMECs) were purchased from ATCC. HMECs were maintained in basal growth medium.

### 2.3. Generation of HLA-E Knockout MCF7 Cells

HLA-E genes were knocked out in MCF7 cells by using the CRISPR-Cas9 KO plasmid according to the manufacturer’s instructions (Santa Cruz Biotechnology). The HLA-E knockout cells were enriched by staining the cells with anti-HLA-E mAb and sorting the HLA-E-negative cells (MCF7ΔHLA-E).

### 2.4. Construction of TCR Expression Vectors

TCR expression vectors were constructed as previously described. Briefly, the TCRβ PCR fragment, the codon-optimized gene encoding the human TCRβ constant-1 region conjugated with the self-cleaving P2A peptide (Cb1-P2A-fragment), the TCRα PCR fragment, and the codon-optimized TCRα constant region gene (Ca-fragment) were assembled in a linearized pMXs-IRES-GFP retroviral vector using the Gibson Assembly Master Mix (New England Biolabs, Ipswich, MA, USA). The constructed plasmid vector, pMXs-TCRβ-P2A-TCRα-IRES-GFP, was used for retrovirus production. For TCR 10-59, we constructed a hybrid expression vector as described previously [15].

### 2.5. Retrovirus Production

Phoenix-A cells were used to produce retrovirus for transducing TCR genes into human peripheral blood mononuclear cells (PBMCs), and Plat-E cells were used to produce retrovirus for transducing TCR genes into mouse lymphocytes. Phoenix-A or Plat-E cells (3.7 × 10^6^) were cultured in a 10 cm dish with 10 mL of DMEM culture medium one day before transfection. TCR expression vectors were transfected into the cells using FuGENE6 transfection reagent (Promega) according to the manufacturer’s instructions. The cells were cultured at 37 °C in a 5% CO_2_ atmosphere. The following day, the cell medium was exchanged. After 2 days of culture, the culture supernatant was harvested, filtered with a 0.22-μm filter (Merck Millipore, Burlington, MA, USA), and stored at −80 °C until use.

### 2.6. Retroviral Transduction of TCRs into Primary T Cells

Human experiments were performed with the approval of the Ethics Committee of the University of Toyama. PBMCs were isolated by Ficoll–Hypaque (Promo cell, Heidelberg, Germany) density gradient centrifugation of peripheral blood from healthy donors. For retroviral TCR transduction, 1 × 10^6^ PBMCs were stimulated with CD3/CD28 Dynabeads (Invitrogen, Dynal AS, Oslo, Norway) in the presence of human IL-2 (PeproTech, Rocky Hill, NJ, USA) for two days. The PBMCs were harvested and resuspended at 5 × 10^5^ cells mL^−1^ in RPMI culture medium in the presence of human IL-2 (30 U mL^−1^). Meanwhile, the wells of 24-well plates were coated with 0.3 mL of RetroNectin (50 μg mL^−1^, Takara, Kyoto, Japan) at 4 °C overnight. TCR-encoding retroviruses were adhered to the wells by centrifugation for 2 h at 1900× *g* and 32 °C. The stimulated PBMCs were then added to the retrovirus-loaded wells, centrifuged at 1000× *g* for 10 min at 32 °C, and incubated overnight at 37 °C in a 5% CO_2_ atmosphere_._ The PBMCs were transferred onto freshly prepared retroviral-coated plates and incubated at 37 °C in 5% CO_2_. The following day, TCR-transduced PBMCs were expanded in the presence of human IL-2 (30 U mL^−1^) for two more days. The TCR-transduced PBMCs were harvested and stored at −80 ℃ until use.

All protocols for the animal experiments were approved by the Committee on Animal Experiments of the University of Toyama. For TCR transduction into mouse splenic T cells, splenocytes were stimulated in vitro with mouse T-Activator CD3CD28 Dynabeads (Invitrogen) in the presence of recombinant mouse IL-2 (PeproTech) for two days. Human TCR-encoding retroviruses were used to infect stimulated splenocytes as described for the retroviral transduction of TCRs into human PBMCs.

### 2.7. Tetramer Staining of BW hCD8⍺β+ Cells Expressing HLA Class-I-Unrestricted TCRs

Approximately 2 × 10^6^ BW-hCD8⍺β+ cells expressing various HLA class-I-unrestricted TCRs were stained with APC-conjugated human tetramers of MR1-5-OP-RU or MR1- Ac-6-FP for 20 min in the dark at room temperature. The cells were then washed three times with PBS containing 2% FCS (PBS/FCS) and resuspended in 200 μL PBS/FCS. Cells were also stained with APC-Cy7-conjugated anti-mouse CD3 mAb for 20 min on ice. The cells were again washed three times with PBS/FCS. The cells were then resuspended in PBS/FCS. Data were acquired using a FACSCanto (Becton Dickinson) and analyzed with FlowJo (Version 10.6.2) software.

### 2.8. RNA Extraction and RT-PCR

RNA was extracted from MCF7WT and MCF7ΔMR1 by using a Nucloespin RNA Mini Kit (MACHERY-NAGEL, Duren, Germany), and cDNA was synthesized by using PrimeScript Reverse Transcriptase (Takara Bio). MR1-encoding cDNA was then amplified by using the following primers: 5′-AGGGGTTACAGCTCTCTTCTG-3′—MR1 forward; 5′-TTGATGCCCACGCCTG-3′—MR1 reverse. Primers for glyceraldehyde 3-phosphate dehydrogenase (GAPDH) were used as a control: 5′-GTCTCCTCTGACTTCAA-3—GAPDH forward; 5′-ACCACCCTGTTGCTGTA-3′—GAPDH reverse. Products from PCR were then separated by agarose gel electrophoresis, and MR1 deletion was confirmed (Appendix A).

### 2.9. Detection of MR1 Surface Expression

MCF7 cells (10^5^) were incubated overnight with 5-OP-RU, Ac-6-FP, or medium alone. The cells were washed and stained with the APC-MR1 mAb (clone 26.5) on ice for 20 min and washed twice with PBS. The cells were suspended in PBS. Data were acquired by using a FACSCanto and analyzed with FlowJo software. Flow cytometric analysis adhered to the guidelines [16].

### 2.10. Detection of HLA-E and CD1s Expression

MCF7 cells were stimulated with 100 U mL^−1^ hIFN-γ for 48 h. The cells were washed and stained with an APC-HLA-E, CD1a, CD1b, CD1c, or CD1d mAb on ice for 20 min and washed twice with PBS. The cells were suspended in PBS. Data were acquired by using a FACSCanto and analyzed with FlowJo software.

### 2.11. ELISA of Secreted Human IFN-γ

TCR gene-transduced PBMCs (1 × 10^5^) were cocultured with 1 × 10^5^ human IFN-γ-treated MCF7 cells in 200 µL RPMI 1640 culture medium in a 96-well plate. Where indicated, the cells were cocultured with different E:T ratios. After 24 h of culture, the supernatants were harvested, and the IFN-γ-production was measured by ELISA (R&D Systems, Minneapolis, MN, USA) according to the manufacturer’s instructions.

### 2.12. ELISA to Evaluate the Stimulation of TCR-Expressing BW Cells

TCR gene-transduced BW-hCD8⍺β+ or BW-hCD8⍺β- cells (1 × 10^5^) were cocultured with 1 × 10^5^ human IFN-γ-treated MCF7 cells in 200 µL DMEM culture medium in the presence of 100 U mL^−1^ mouse IL-1α in a 96-well plate. Where indicated, 5-OP-RU, Ac-6-FP, or anti-MR1 mAb was added to the culture at the indicated concentrations. After 24 h of culture, the supernatants were harvested, and the IL-2-production was measured by ELISA (R&D Systems) according to the manufacturer’s instructions.

### 2.13. Overexpression of MR1 in MCF7 Cells

The cDNA of MR1 genes was transferred into a piggyBac single promoter vector containing IRES-GFP (System Biosciences, Palo Alto, CA, USA). The resultant vector was transfected into MCF7 cells using the super piggyBac transposase expression vector (System Biosciences). To select the MR1-expressing MCF7 cells (MCF7WT-MR1++ and MCF7ΔMR1-MR1++), the cells were stained with APC-conjugated anti-human/mouse/rat MR1 mAb, and the MR1-expressing cells were enriched by FACSAria II (Becton Dickinson).

### 2.14. Generation of K43A-Mutated MR1-Expressing MCF7 Cells

The construction of the K43A-substituted human MR1 expression vector was outsourced to VectorBuilder (Chicago, IL, USA), and the vector was transfected into MCF7ΔMR1 cells. The MR1^K43A^ molecules in the cells were stained with APC-conjugated anti-human/mouse/rat MR1 mAb, and the MR1^K43A^-expressing cells were enriched by FACSAria II.

### 2.15. Cytotoxicity Assay

For the cytotoxicity assay, we cocultured TCR-expressing PBMCs with MCF7-Luc cells (1 × 10^4^) in 96-well plates at the indicated effector-to-target (E/T) ratios and incubated them for 24 h. Cytotoxicity against MCF7-Luc cells was assessed by measuring the cell-associated luciferase activity using the Steady-Glo luciferase assay system (Promega), as described previously [17].

### 2.16. Statistical Analysis

The data are presented as the means ± S.Ds, with statistically significant differences determined by the tests indicated in the figure legends. All statistical analyses were performed using GraphPad Prism 9 (Version 9.0.2, La Jolla, CA, USA) and Microsoft Excel 2016 software (Version 16.89.1).

## 3. Results

### 3.1. MR1/5-OP-RU Tetramer Binding

As previously reported, we obtained 21 HLA-independent TCRs from CD8^+^ TILs from two breast cancer patients. Preliminary experiments indicated that 15 of these TCRs reacted to MCF7 cells in an MR1-dependent manner when expressed in BW-hCD8⍺β+ (BW) cells [15]. To investigate the MR1 reactivity of these TCRs, we stained TCR-expressing BW cells with human MR1 (hMR1) tetramers. No binding of the TCRs with the hMR1 tetramers (neither hMR1/5-OP-RU nor hMR1/6-FP) was observed, except for TCR2-78 (Figure 1 and Appendix A, gating strategy shown in Appendix A). Of the 15 HLA-independent TCRs, only TCR2-78 was composed of TRAV1-2–TRAJ33, and TRBV6-1 like MAIT TCR (Table 1).

### 3.2. Characterization of Unconventional HLA-Class I-like Molecules in Various Cell Lines (MR1, CD1, and HLA-E)

To characterize the antigen-presenting molecules of the HLA-independent TCRs, we compared the expression of HLA-class I-like molecules (HLA-E, CD1, and MR1) across various cell lines (Gating strategy shown in Appendix A). MCF7 cells expressed HLA class I (HLA-I) molecules on the cell surface but did not express the HLA-E, CD1, and MR1 molecules (Appendix A). HLA-E expression in tumor cells can be induced by IFN-γ stimulation [18]. When MCF7 cells were stimulated with IFN-γ, the cell surface expression of the HLA-E molecule was induced (Appendix A). In contrast, MR1 expression was not observed in MCF7 cells after treating them with IFN-γ. MR1 expression can be induced by the MR1 ligand Ac-6-FP [19]. Eckle et al. showed that MR1 was strongly upregulated by Ac-6-FP as compared to 5-OP-RU. We incubated MCF7 cells overnight with MR1 ligands 5-OP-RU or Ac-6-FP and observed that the expression of the MR1 molecule in wild-type MCF7 cells (MCF7WT) was strongly upregulated by Ac-6-Fp and weakly upregulated by 5-OP-RU (Appendix A). Our results correspond to those described by Eckle et al. [20]. MR1 upregulation by Ac-6-FP was not observed in MCF7Δβ2m or MCF7ΔMR1 cells (Appendix A). MR1 expression was also observed on the cell surface of the other breast cancer cell lines (Table 2 and Appendix A) and colon, lung, myeloid, and lymphoid cancer cell lines (Table 2 and Appendix A) when these cells were treated with Ac-6-Fp. The expressions of the HLA-E and CD1 molecules in the other cell lines were also examined (Table 3 and Table 4 and Appendix A).

### 3.3. Reactivity of HLA-Unrestricted TCR-Expressing Mouse Spleen T Cells and Human PBMCs to MCF7 Cells

We then examined the reactivity of these TCRs by expressing them in mouse T cells or human PBMCs. To this end, we selected four MR1-restricted TCRs (TCR2-78, TCR2-81, TCR10-59, and TCR10-69), which reacted to MCF7 cells in a manner independent of the patient’s HLA when expressed in BW cells [15]. We also used TCR2-15, whose reactivity was dependent on the patient’s HLA-B*59:01 as a representative of HLA class-I restricted TCRs and analyzed the response to MCF7 cells expressing HLA-B*59:01. Although all these TCRs reacted to MCF7 cells when expressed in mouse T cells and secreted mouse IFN-γ (Figure 2A), HLA-independent TCRs (2-78, 2-81, and 10-69), other than TCR10-59, unexpectedly did not react to MCF7 cells when expressed in human PBMCs from a healthy donor (Figure 2B). This may be due to the incompatibility of TCR2-78, 2-81, and 10-69 with endogenous TCRs. To exclude the possibility of alloreactivity, we transduced these TCRs into PBMCs from another donor and observed corresponding results (Figure 2A and Appendix A). The response of the HLA-independent TCRs (2-78, 2-81, 10-59, and 10-69) expressed in mouse T cells was weaker than the response of TCR10-59 expressed in human PBMCs. To analyze the cytotoxicity to MCF7 cells, we used TCR10-59-expressing human PBMCs. Thus, we focused on TCR10-59 for further detailed analysis.

### 3.4. Reactivity of TCR10-59 to MCF7 KO Cells

To clarify the antigen-presenting molecules for HLA-independent TCR10-59, we first knocked out the HLA-I A, B, and C (MCF7ΔHLA-I), β2-microglobulin (MCF7Δβ2m) and MR1 (MCF7ΔMR1) genes in MCF7 cells by using the CRISPR/Cas9 [15]. TCR10-59 showed reactivity against MCF7 cells in an HLA-independent manner by secreting TNF-⍺ or IFN- γ, while conventional TCR2-15 responded to MCF7 cells in an HLA-dependent manner. (Figure 3A,B). TCR10-59 did not respond to MCF7 cells that lacked the β2-microglobulin or MR1 gene. To confirm the MR1-dependent reactivity of TCR10-59, we overexpressed MR1 molecules in MCF7WT (MCF7WT-MR1^++^) cells or MCF7ΔMR1 cells (MCF7ΔMR1-MR1^++^)**.** MR1 molecules were detected on their cell surface without incubation with Ac-6-FP (Appendix A). As shown in Figure 3C, TCR10-59 and TCR2-78 responded more strongly to MCF7WT-MR1^++^ cells or MCF7ΔMR1-MR1^++^ cells than MCF7WT cells. Next, we knocked out the HLA-E gene in MCF7 cells by using the CRISPR/Cas9 system to generate MCF7ΔHLA-E cells. HLA-E deletion was confirmed by staining the cells with anti-HLA-E mAb (Appendix A). Functional analysis of TCR10-59 showed that it reacted against MCF7ΔHLA-E cells (Appendix A) but not against MCF7ΔMR1 cells, which further authenticates our data. Next, we analyzed the cytotoxicity of TCR10-59-transduced PBMCs against MCF7 cells expressing luciferase in vitro and found that TCR10-59-transduced T cells killed MCF7 cells in an HLA-independent manner, while no cytotoxicity was observed against MCF7ΔMR1 cells (Figure 3D). These data show that TCR10-59 reacts to MCF7 cells in an HLA-independent and MR1-restricted manner.

### 3.5. Reactivity of TCR10-59 to Breast Cancer Cells but Not to Other Cancer Cells

Recently, MR1-restricted T cells were discovered that were reactive to a group of human cancer cell lines [12]. We examined the target cells for TCR10-59. TCR10-59-expressing human PBMCs reacted not only to MCF7 cells but also to MDA-MB-231 breast cancer cells but did not react to other cancer cell lines (Colo205 colon cancer cells, A549 lung cancer cells, K562 myeloid cells, and Karpas 299 lymphoid cells) (Figure 4A and Appendix A). They did not react to another breast cancer cell line, ZR-75-1, although it showed a higher surface expression of MR1 (Appendix A). TCR2-78, TCR2-81, and TCR10-69 also reacted to MCF7 cells but did not react to MDA-MB-231 cells when expressed in mouse T lymphocytes (Appendix A). This may be due to the low reactivity of these TCRs compared to that of TCR10-59, or they recognized different antigens from TCR10-59. To examine whether those non-stimulatory cancer cell lines could stimulate MAIT cells, we examined the response of TCR2-78 to 5-OP-RU, which has a typical MAIT TCR sequence. As shown in Appendix A, the TCR2-78-expressing BW cells responded to 5-OP-RU presented by MCF7 cells. Similarly, the TCR2-78-expressing BW cells responded to 5-OP-RU presented by the non-stimulatory cancer cell lines (MDA-MB-231, ZR-75-1, A549, and K562). 5-OP-RU is presented in the context of MR1 molecules, because 5-OP-RU failed to stimulate TCR2-78 when presented by MCF7ΔMR1 cells (Appendix A). Thus, MR1-restricted TCRs derived from the TILs of breast cancer patients reacted to an antigen expressed in MCF7 or MDA-MB-231 breast cancer cells but not expressed in other cancer cell lines, including the ZR-75-1 breast cancer cell line. Another possibility is that the antigen presentation in those non-stimulatory cancer cells is inefficient compared to the MCF7 cells or MDA-MB-231 cells. We then assessed whether TCR10-59 reacted to normal breast cells. For this purpose, we used human mammary epithelial cells (HMECs). TCR10-59 demonstrated reactivity toward MCF7 breast cancer cells but not toward normal breast cells (Figure 4B).

Recently, Souter et al. showed that CD8-engagement of MR1 enhanced the antigen responsiveness of MAIT and MR1-reactive T cells [21]. To examine the involvement of CD8 in our MR1-restricted TCR responses, we expressed TCR10-59 in hCD8⍺β− BW cells. In the absence of hCD8 expression in BW cells, the reactivity of TCR10-59 was reduced. However, TCR10-59-expressing BW cells significantly responded to MCF7 cells even in the absence of hCD8 (Appendix A).

### 3.6. Ac-6-FP and MR1 mAb Did Not Inhibit the Reactivity of MR1-Restricted T Cells to MCF7 Cells

We next analyzed the effects of known microbial antigens in the context of MR1; 5-OP-RU and Ac-6-FP [22]. For this purpose, we cocultured BW cells expressing TCR2-78 (a typical MAIT TCR) or TCR10-59 (an MR1-restricted TCR) with IFN-γ-stimulated MCF7 cells, along with 5-OP-RU, Ac-6-FP, and anti-MR1-mAb overnight in a dose-dependent manner. The next day, we measured IL-2-secretion by the BW cells. We found that 5-OP-RU enhanced IL-2-production from typical MAIT TCR (2-78)-expressing BW cells, while no effect was observed on MR1-restricted TCR (10-59)-expressing BW cells (Figure 5A). Additionally, Ac-6-FP, a non-stimulatory ligand for MR1, substantially suppressed IL-2-secretion in response to the agonist ligand (5-OP-RU) (Figure 5B). Similarly, anti-MR1-mAb also inhibited the effect of the agonist ligand (5-OP-RU) (Figure 5C). TCR 2-78- and TCR10-59-expressing BW cells responded to MCF7 cells in the absence of 5-OP-RU, and this response was not inhibited by Ac-6-FP or anti-MR1-mAb (Figure 5B,C). Although the addition of Ac-6-FP increased the expression of MR1 in breast cancer cells, the addition of Ac-6-FP to the breast cancer cells did not affect the response of TCR10-59 (Appendix A). Additionally, we examined the effect of folic acid in the medium. We added an excessive amount of folic acid (up to 100 μg/mL) to the reaction of TCR2-78 and TCR10-59 against MCF7 cells (the DMEM contained 4 μg/mL folic acid). The results show that the reactivity of the TCR-expressing BW cells to MCF7 cells was not affected by folic acid in the medium (Appendix A).

### 3.7. Reactivity to Breast Cancer Cells Is Dependent on K43 Residue of MR1 Molecules

K43 residue in MR1 molecules was shown to be important for the presentation of their ligands to previously reported MR1-restricted TCRs, including those for MAIT cells [23]. In detail, MR1 ligands form a Schiff base with the K43 residue of MR1. This formation induces the folding of MR1 and its exit from endoplasmic reticulum. To examine the role of the K43 residue of MR1 in the presentation of antigenic molecules for TCR10-59, we analyzed the response of TCR10-59 to K43A-MR1-expressing MCF7 cells [24]. To analyze the role of K43 in MR1 in the context of the reactivity of TCR10-59, we expressed wild-type MR1 and MR1 with K43A substitution in MCF7ΔMR1 cells (MCF7ΔMR1-MR1^++^ and MCF7ΔMR1-MR1^K43A^, respectively). The MR1 expression levels in MCF7ΔMR1-MR1^++^ and MCF7ΔMR1-MR1^K43A^ were much higher than that in MCF7WT (Appendix A). The MR1 expression level in MCF7ΔMR1-MR1^K43A^ cells was higher than that of MCF7ΔMR1-MR1^++^ cells, which may be because the folding of K43A-mutant MR1 and its expression is independent of ligand binding [6].

We cocultured TCR2-78- and TCR10-59-expressing BW cells with IFN-γ-stimulated MCF7WT, MCF7ΔMR1-MR1^++^, and MCF7ΔMR1-MR1^K43A^ cells. After overnight culture, we measured the IL-2 secretion by the BW cells. The response of TCR2-78 to the MCF7ΔMR1-MR1^K43A^ cells was significantly reduced, and the response to the MCF7ΔMR1-MR1^++^ cells was significantly enhanced compared to that to the MCF7WT cells (Figure 5D). These responses were further boosted by the addition of 5-OP-RU. Similarly, the response of TCR10-59 to the MCF7-MR1^K43A^ cells was significantly reduced, and the response to the MCF7ΔMR1-MR1^++^ cells was significantly enhanced compared to that to the MCF7WT cells, although the addition of 5-OP-RU failed to augment the reactivities (Figure 5D).

## 4. Discussion

MR1 molecules present antigens derived from the vitamin B2 (riboflavin) metabolic pathway to MAIT cells [22]. Recently, self-reactive or cancer-reactive MR1-restricted T cells were identified in the blood of healthy individuals, but their antigens remained unknown [11,12]. Later, a cancer-reactive TCR [12] was shown to be restricted to an MR1 allele (MR1*04) and was not pan-cancer-reactive [25]. In this study, we analyzed MR1-restricted TCRs obtained from the TILs of two breast cancer patients [15]. We demonstrated that our TCRs reacted to breast cancer cells but not to the other cancer cells nor normal breast tissue cells, while MR1 T cells reported by Crowther et al. reacted to various tumor cell lines [12], and MR1 T cells reported by Lepore et al. reacted not only to MR1-expressing leukemia cells but also MR1-expressing melanoma cells [11]. Furthermore, the reactivity of our MR1-restricted TCRs to breast cancer cells was not inhibited by MR1 antagonist, Ac-6-FP, or anti-MR1 mAb, while that of MAIT cells and the reported MR1-restricted T cells was inhibited by 6-FP, Ac-6-FP, and anti-MR1 mAb (Appendix A) [11,12]. One possible explanation of our results is that MR1 can present an antigen produced in some breast cancer cells and MR1-restricted TCR-expressing T cells infiltrate into tumors in antigen-dependent manner.

The MR1 heavy chain has structural homology with HLA-A2 (39% sequence identity) and CD1d (22% sequence identity) and binds the ligand 6-FP in its cleft by forming a Schiff base between the ligand and K43 residue of MR1 [6,26]. Concerning the involvement of the K43 residue of MR1 in antigen recognition, Lepore et al. reported that the reactivity of their MR1-restricted T cells was not dependent on the K43 residue in MR1 [11], but Crowther et al. reported that the reactivity of their MR1-restricted T cells was dependent on the K43 residue of MR1 [12], while the reactivity of both T cells was inhibited by 6-FP or Ac-6-FP and by anti-MR1 mAb (Appendix A). Since reactivity of our MR1-restricted TCR (10-59) to breast cancer cells was not inhibited by Ac-6-FP, but the response was significantly reduced by the K43A mutation of MR1, it is speculated that the antigenic molecule bound to MR1 more strongly than Ac-6-FP, and its binding was affected by the K43A mutation of MR1 molecules.

Although MR1 has been identified as a monomorphic antigen-presenting molecule, Rozemuller et al. recently reported that MR1 has at least six alleles grouped with coding region alterations [27]. Howson et al. previously showed that a patient who had a homozygous mutation in MR1*04 (R9H) had a history of difficult-to-treat viral and bacterial infections, and the R9H MR1 protein could not bind to the microbial MR1 ligand 5-OP-RU [28]. It is possible that the difference in MR1 alleles might affect the antigen presentation of MR1 molecules. Our TCR10-59 reacted to MCF7 and MDA-MB-231 breast cancer cells but not to ZR751 breast cancer cells. Public databases (CCLE and Crown Bioscience) (https://sites.broadinstitute.org/ccle/datasets accessed on 3 April 2023) show that the MR1 alleles of MCF7, MDA-MB-231, and ZR751 are *02, *01, and *01 respectively, suggesting that the reaction of TCR10-59 to these three cell lines was independent of the MR1 alleles.

The present study has several limitations. First, we did not identify the antigen of our MR1-restricted TCRs. Our MR1-restricted TCRs reacted to MCF7 cells and MDA-MB-231 breast cancer cells but did not react to other cancer cell lines. Therefore, we are trying to identify the antigenic molecules of our TCRs by comparing the molecules in MCF7 cells, MDA-MB-231 cells, and other cell lines. Second, we have not determined the MR1 alleles of the cell lines ourselves. So, we will determine the MR1 alleles of the cell lines in the future. Third, we only analyzed TCRs derived from the TILs of two breast cancer patients. It may be important to analyze TCRs derived from TILs of other cancer patients.

## 5. Conclusions

We have reanalyzed the MR1-restricted TCRs that reacted to breast cancer cells and shown that they responded to some breast cancer cell lines but not to other cancer cell lines. The identification of the antigen for our MR1-restricted TCRs will expand our understanding of the physiological role of MR1 and MR1-restricted T cells. To this end, we plan to perform a (1) genome-wide knockout experiment using the CRISPR/Cas9 system [12,29] and (2) a cDNA library experiment to identify the molecules that are involved in the production of antigenic molecules and their presentation in MR1 molecules.

## Figures and Tables

**Figure 1 cells-13-01711-f001:**
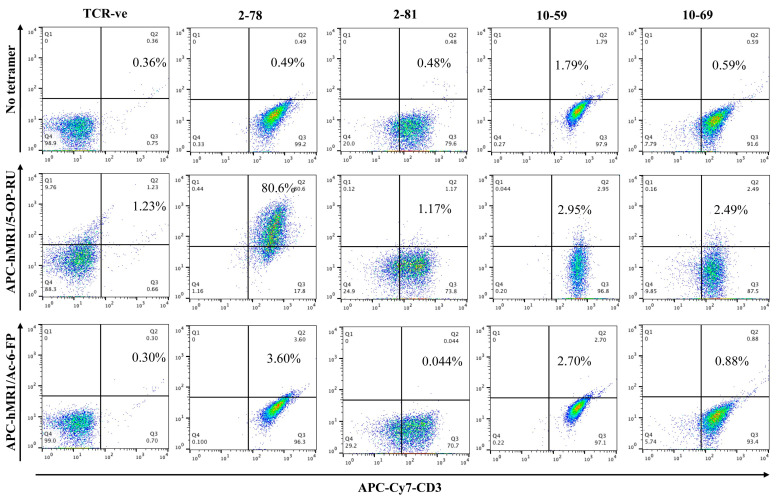
MR1 tetramer staining. BW-hCD8⍺β+ cells transfected with MR1-restricted TCRs were stained with APC-conjugated MR1 tetramers (hMR1/6-FP or hMR1/5-OP-RU) and APC Cy-7 conjugated anti-mouse CD3 antibody. Right upper quadrant is showing CD3+Tetramer+ TCR-expressing BW-hCD8⍺β+ cells. TCR-negative (TCR-ve) BW-hCD8⍺β+ cells were used as a negative control. The plots are representative of two independent experiments.

**Figure 2 cells-13-01711-f002:**
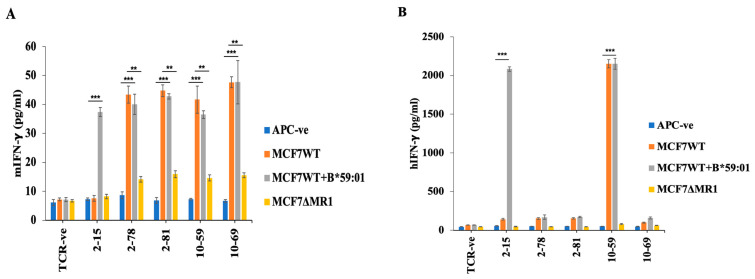
The reactivity of TCR-transduced mouse splenic T cells and human PBMCs. (**A**) TCR-expressing mouse splenic T cells were cocultured with IFN-γ-stimulated MCF7 cells. The reactivity against the MCF7 cells was measured by mouse IFN-γ ELISA in triplicate. TCR-negative (TCR-ve) T cells were used as a negative control. (A single experiment is representative of three independent experiments). (**B**) Human PBMCs from healthy donor A were transduced with TCRs and cocultured with IFN-γ-stimulated MCF7 cells. IFN-γ-production in the supernatant was measured by ELISA in triplicate. TCR-negative (TCR-ve) T cells were used as a negative control. A single experiment is representative of 3 independent experiments. The mean and SD values from technical triplicate cultures are indicated. Unpaired, two-tailed t-tests were performed (** *p* <0.01 and *** *p* <0.001).

**Figure 3 cells-13-01711-f003:**
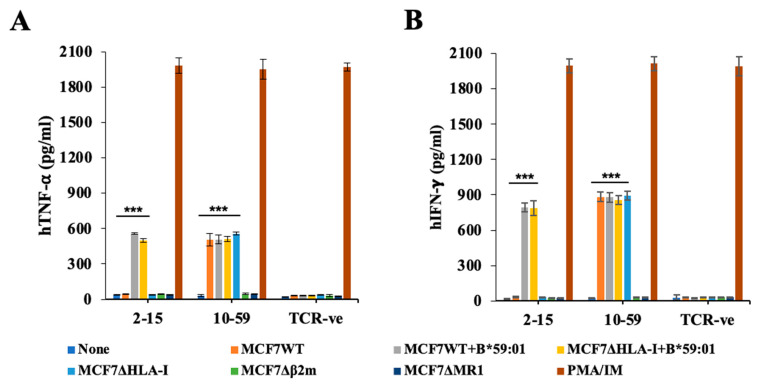
The reactivity of TCR10-59 to MCF7 WT and various KO cells. (**A**,**B**) TCR10-59 and TCR2-15 were transduced into human PBMCs from healthy individuals, and the cells were cocultured with IFN-γ-stimulated MCF7 cells. TNF-⍺ (**A**) and IFN-γ (**B**) production in the supernatant was measured by ELISA. TCR-negative (TCR-ve) T cells were used as a negative control. T cells stimulated with PMA + ionomycin were used as a positive control. (A single experiment is representative of three independent experiments). (**C**) TCR2-78- or TCR10-59-expressing BW-hCD8⍺β+ cells were cocultured with MR1-overexpressing MCF7WT cells or MCF7ΔMR1 cells, and mouse IL-2-secretion was analyzed by ELISA. The mean and SD values from technical triplicate cultures are indicated. (**D**) PBMCs were retrovirally transduced with conventional (TCR2-15) or unconventional (TCR10-59) TCR genes and cocultured with luciferase-expressing IFN-γ-stimulated MCF7WT, MCF7 WT+B*59:01, and MR1-deleted MCF7 cells at different effector-to-target (E:T) ratios. TCR-negative (TCR-ve) T cells were used as a negative control. The cytotoxic effects were observed by identifying the viable cells by luciferase activity. A single experiment is representative of three independent experiments with similar results. The mean and SD values from technical triplicate cultures are indicated. Unpaired, two-tailed t-tests were performed (** *p* < 0.01 and *** *p* < 0.001).

**Figure 4 cells-13-01711-f004:**
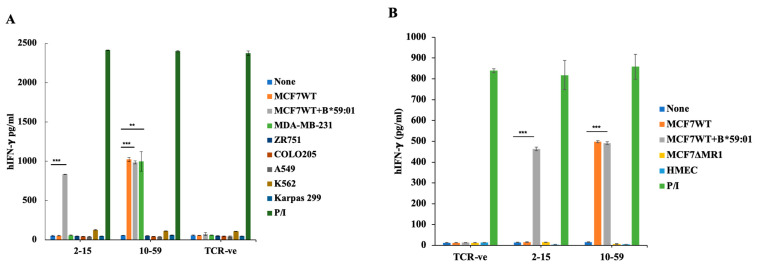
The breast cancer cell-specific reactivity of TCR10-59. (**A**) TCR 2-15- and TCR10-59-expressing PBMCs were cocultured with various IFN-γ-stimulated human cancer cell lines (MDA-MB-231 and ZR-75-1 breast cancer cells, Colo205 colon cancer cells, A549 lung cancer cells, K562 myeloid cells, and Karpas 299 lymphoid cells) for 24 h. TCR-untransduced PBMCs were used as a negative control. T cells stimulated with PMA + ionomycin were used as a positive control. ELISA was performed to measure IFN-γ secretion in the supernatants. The mean and SD values from technical triplicate cultures are indicated. (**B**) TCR-transduced human PBMCs from healthy individuals were cocultured with IFN-γ-stimulated MCF7 cells as well as normal breast cells (HMECs). IFN-γ production in the supernatant was measured by ELISA. TCR-negative (TCR-ve) T cells were used as a negative control. For the positive control, T cells stimulated with PMA and ionomycin (P/I) were used. (A single experiment is representative of 2 independent experiments). The mean and SD values from technical triplicate cultures are indicated. Unpaired, two-tailed *t*-tests were performed (** *p* < 0.01 and *** *p* < 0.001).

**Figure 5 cells-13-01711-f005:**
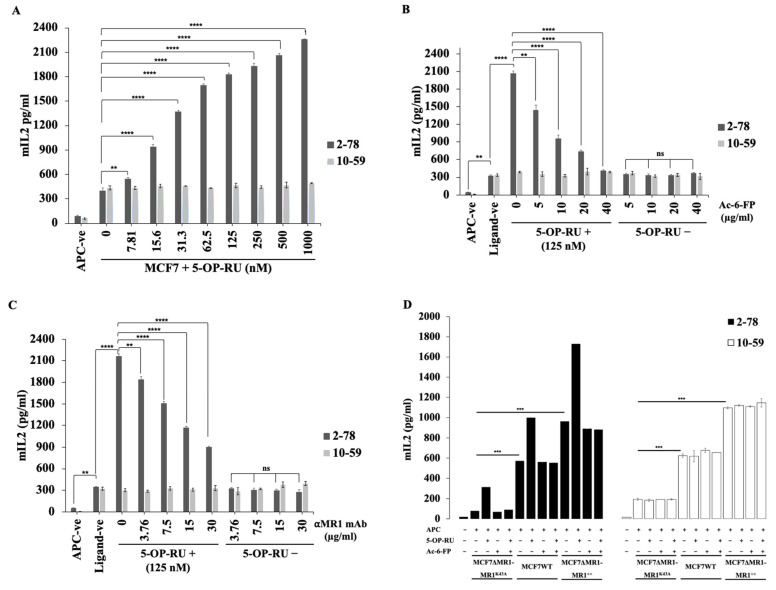
The effects of known microbial MR1 antigens and dependency on K43 MR1-restricted TCRs. TCR2-78- and TCR10-59-expressing BW cells were cocultured with IFN-γ-stimulated MCF7 cells. For stimulation, 5-OP-RU (**A**), and for inhibition, Ac-6-FP (**B**) and anti-MR1 antibody (**C**) were added to the cultures at the indicated concentrations. The dose-dependent responses were assessed by analyzing the IL-2 production in the supernatant. The experiment was performed three times with similar results. (**D**) TCR2-78- and TCR10-59-expressing BW cells were cocultured with IFN-γ-stimulated MCF7WT, MCF7ΔMR1-MR1^K43A^, and MCF7ΔMR1-MR1^++^ cells. 5-OP-RU (125 nM) and Ac-6-FP (40 μg mL^−1^) were added to the cultures for the stimulation and inhibition of TCR-expressing BW cells, respectively. The IL-2 production in the supernatant was measured as the assessment of responses. The mean and SD values from technical triplicate cultures are indicated. Unpaired, two-tailed t tests were performed (** *p* < 0.01, *** *p* < 0.001, **** *p* < 0.0001 and ns—not significant).

**Table 1 cells-13-01711-t001:** TCR sequences of HLA-independent TCRs.

TCR	TRAV CDR3ɑ TRAJ	TRBV CDR3β TRAJ
2-78	1-2 CAAIDSNYQLIW 33	6-1 CASKERSGSGDGEQYF 2-7
2-81	19 CALPSRLMF 31	2 CASSLTSIYEQFF 2-1
10-59	26-1 HRQSRSR#YGGSQGNLIF 42	5-4 CASSFYGSETQYF 2-5
10-69	13-1 CAASMGNTPLVF 29	20-1 CSARVEKLFF 1-4

**Table 2 cells-13-01711-t002:** MR1 expression in human cancer cell lines after pulsing with Ac-6-FP.

Cells	Mean Fluorescence Intensity		
	Isotype	Without Ac-6-FP	With Ac-6-FP
MCF7	4.7	5.4	20.4
MDA-MB-231	3.2	3.7	6.7
MDA-MB-453	2.8	4.1	30.1
MDA-MB-468	2.8	5.1	10.0
BT474	3.9	4.6	8.9
ZR-75-1	2.6	4.8	34.5
A549	2.6	4.5	5.7
COLO205	2.0	2.6	12.2
Jurkat	5.2	12.9	84.0
Karpas299	4.3	7.5	39.1
K562	3.1	6.8	51.1

**Table 3 cells-13-01711-t003:** HLA-E expression in human cancer cell lines after treatment with human IFN-γ.

Cells	Mean Fluorescence Intensity		
	Control	IFN-γ -	IFN-γ +
MCF7	5.23	6.28	62.9
MDA-MB-231	5.09	6.09	8.23
MDA-MB-453	15.2	18.9	51.3
MDA-MB-468	13.6	14.8	15.2
BT474	10.2	12.1	14.4
ZR-75-1	7.00	8.66	45.5
A549	5.5	11.6	25.6
COLO205	3.5	22.0	86.4
Jurkat	3.4	4.0	15.8
Karpas299	3.63	18.5	15.1
K562	5.63	7.93	9.84

**Table 4 cells-13-01711-t004:** Expression of CD1 molecules in human cancer cell lines after treatment with human IFN-γ.

Cells	Mean Fluorescence Intensity				
	Control	CD1a	CD1b	CD1c	CD1d
MCF7	6.61	7.68	5.74	6.23	5.92
MDA-MB-231	5.61	5.63	5.34	5.87	5.62
MDA-MB-453	8.50	8.53	8.49	8.42	7.71
MDA-MB-468	8.60	8.65	8.11	8.50	9.94
BT474	11.0	11.3	9.81	10.4	11.4
ZR-75-1	11.4	11.6	11.3	12.0	14.4
A549	11.8	13.4	12.0	11.4	8.01
COLO205	8.30	9.34	8.42	8.63	9.28
Jurkat	8.0	440	231	366	916
Karpas299	3.63	3.81	3.96	3.78	13.5
K562	5.63	7.13	4.53	4.72	4.22

## Data Availability

The data that support the findings of this study are available from the corresponding author upon reasonable request.

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
