# Peer review of "Characterization of Tumor-Infiltrating Lymphocyte-Derived Atypical TCRs Recognizing Breast Cancer in an MR1-Dependent Manner"

_cells, 2024, doi:10.3390/cells13201711_

Round 1
Reviewer 1 Report
Comments and Suggestions for Authors
In this manuscript Hayee et al. report the identification of TCRs retrieved from T cells infiltrating breast cancer tumors that are reactive to MR1 molecules but not to currently known antigens presented by MR1. Furthermore, these TCRs are not TRAV1-2+, which is usually associated with MAIT cells. The results are usually clearly presented and controlled. However, a table with the information regarding these TCRs (V-J chains and CDR3 sequences) is warranted so that the results can be independently reproduced. Furthermore, the authors rely on database information for the identification of MR1 alleles expressed by the various cell line used in the study. The sequence and allele of MR1 molecules for each of the cell lines should be independently verified by the authors. Finally, have the authors tried to generate extracts from the MCF7 cell line and get these extracts to be presented by another breast cancer-derived MR1-expressing cell line (ZR-75-1) that do not express the putative antigen(s) to the TCR (10-59) expressing BW cells? A positive results in such experiment would strongly support the author’s claims for the existence of breast cancer specific MR1-presented antigens.
Comments on the Quality of English LanguageEnglish language was ok but could be improved in some places. Large language models, such as chatGPT could be useful for the fluidity of the language, assuming that the authors meticulously check for the veracity of the statements.
Author Response
Point-by-point response to Comments and Suggestions for Authors |
Comments and Suggestions for Authors In this manuscript Hayee et al. report the identification of TCRs retrieved from T cells infiltrating breast cancer tumors that are reactive to MR1 molecules but not to currently known antigens presented by MR1. Furthermore, these TCRs are not TRAV1-2+, which is usually associated with MAIT cells. The results are usually clearly presented and controlled.
Comment 1: However, a table with the information regarding these TCRs (V-J chains and CDR3 sequences) is warranted so that the results can be independently reproduced.
|
Response 1: Thank you for suggestion. According to the reviewer’s suggestion, we have added the table (Table 1) showing the TCRs (V-J chains and CDR3 sequences).
|
Comment 2: Furthermore, the authors rely on database information for the identification of MR1 alleles expressed by the various cell line used in the study. The sequence and allele of MR1 molecules for each of the cell lines should be independently verified by the authors.
|
Response 2: Thank you very much for suggesting an important point. As suggested by the reviewer, it is important to analyze the MR1 alleles that TCR10-59 reacts to. We tried to determine the sequence of MR1 alleles for our cell lines according to methods from (T. V Cornforth et al. bioRxiv, p. 2023.07.17.548997, Gozalbo-López B et al., Histol Histopathol, 2009 and J. M Greene et al. Mucosal Immunol. 2017) but could not get the sequence data at this time. We apologize that we cannot provide the data regarding the sequence and alleles of MR1 molecules for cell lines we have used in this manuscript. We have also mentioned this limitation in revised manuscript (lines 707-709). We will determine these sequences and alleles of MR1 molecules in the future.
Comment 3: Finally, have the authors tried to generate extracts from the MCF7 cell line and get these extracts to be presented by another breast cancer-derived MR1-expressing cell line (ZR-75-1) that do not express the putative antigen(s) to the TCR (10-59) expressing BW cells? A positive results in such experiment would strongly support the author’s claims for the existence of breast cancer specific MR1-presented antigens.
Response 3: Thank you very much for suggesting this. As suggested by the reviewer, we prepared the extract of MCF7 cells by freezing and thawing method. We then cultured TCR10-59 expressing BW cells together with the other cell line (ZR-75-1) in the presence or absence of MCF7 cell extract. No effect of extract was observed on TCR10-59 stimulation against ZR-75-1 cells. We have not shown the data in the manuscript. We are now trying to analyze metabolite profile of the cell lines. We hope that we will determine the antigenic metabolite in the future.
|
Extract from MCF7 cells did not affect stimulation of TCR10-59. TCR10-59 expressing BW -hCD8⍺β+ cells were co-cultured with IFN-γ-stimulated ZR-75-1 cells in presence or absence of extract of MCF7 cells. MCF7 cells were used as positive control. Mouse IL-2 secretion in the supernatant was measured by ELISA. IFN-γ-stimulated MCF7 cells were used as positive control. Mean and SD values from technical triplicate cultures are indicated.
4. Response to Comments on the Quality of English Language |
Point 1: |
Response 1: We have checked and improved the English language by using AI tool.
|

Reviewer 2 Report
Comments and Suggestions for Authors
In the manuscript entitled “Identification of tumor-infiltrating-lymphocytes-derived atypical TCRs recognizing breast cancer in a MR1-dependent manner”, Hayee et al have reported new data on unconventional T cell receptors (TCR) that recognize tumour cells in a non-HLA restricted manner. This is a continuation of work previously reported in the European Journal of Immunology in 2021 by Yamaguchi et al, where authors used a very efficient technical approach for identifying novel TCRs using recombination and expression in a mouse T cell reporter line (which they call c-FIT). In this previous work, authors identified a number of non-HLA restricted TCRs, which induced T cell responses against MCF7 breast cancer cell line, and seemed to be dependent on MR1 expression. In this paper, authors further characterize the response of these novel TCRs using MR1-associated ligands, and various genetic manipulations of MCF7 HLA or MR1 expression. Authors find that their atypical 10-59 TCR does not bind to MR1-tetramers, responds to breast cancer cells (but not other cancer types), is dependent on MR1 expression for response, not affected by MR1 ligands, and not affected by K43 mutation of MR1. These results seem somewhat contradictory, and thus should be better explained/summarized in the discussions section.
Overall, this work builds nicely on the previous reports about these interesting TCRs, though there are some experimental issues that should be addressed and some editing is required to better contextualize the meaning of these results. Authors are commended on assembling interesting scientific data and a well written report; I look forward to reviewing a revised version of the manuscript soon.
Specific comments:
(1) The title gives the impression that new TCRs are identified here. Perhaps this could be changed to reflect that previously identified TCRs are being characterized in this paper
(2) The abstract is perhaps too specific to those with knowledge of MR1 restricted TCRs. Perhaps better explaining the relevance of AC-6-FP and K43 findings would be helpful to make the impact of the work here more readily apparent.
(3) Given that TCR 10-59 expressing cells seem to be dependent on MR1 expression (as per knockout experiments), but do not bind MR1 tetramers, and do not even react to MR1 modulation via antigen treatment, is a possible explanation that TCR 10-59 could react to a MR1-associated receptor??
(4) Line 249: please provide the specific reference here
(5) Line 270: please clarify how results specifically correlate with the work of Eckle
(6) Line 272: Confirmation of deletion should be described before results with delta-MR1 experiments
(7) Section 3.2: The results discussed here are entirely supplemental. A summary table or figure of this data should be included as a main figure.
(8) Line 265: It appears that with or without IFNg treatment, no expression of MR1 is detected on MCF7 cells, but this is not mentioned in the manuscript. Why is this? Only with AC-6-FP does MR1 expression become apparent on MCF7 cells, and also on other breast cancer cells (S3D), and non-breast cancer cells (S3E)
(9) Given that authors show that in almost all cancer cell lines treatment with Ac-6-FP increases surface MR1 expression, it would be important to test the response of 10-59 TCR expressing cells +/- Ac-6-FP instead of only under IFN-g stimulation. Furthermore, all of the breast cancer cells reported in S3D should tested +/- Ac-6-FP for response to get a better idea of how universal the response on 10-59 is against breast cancer cells. This could be done with either hPBMC or BW clones.
(10)Line 282: The experiment wherein overexpression of patient HLA, HLA-B*59:01 in MCF7 is used should be explained more clearly.
(11)Line 289: What is a “donor B”
(12)Line 290: A better summary of the specific reasoning for focus on TCR10-59 would be helpful (eg, good expression in PBMC and its strong response characteristics)
(13)Line 360: The rationale/background on MR1-K43A should be better introduced to make clear why these experiments were done.
(14)Section 3.8: This could be integrated into section 3.5
(15)Line 385-386: This line does not seem to be needed
(16)In several figures: The control is labeled APC-ve, what does this mean?
(17)Figure 5: In some subfigures the dose is written nM/ml. Is this nmol/mL or nM?
(18)Line 641: While one possible explanation seems to be that MR1 may present a breast cancer antigen, the results are not necessarily suggestive of this. Please reword this sentence.
(19)The conclusion section should be written more generally, and provide some more context for how these results might be applied. A future direction to try to identify the specific antigen for 10-59 is a good idea, but how will this be done?
Author Response
3. Point-by-point response to Comments and Suggestions for Authors |
In the manuscript entitled “Identification of tumor-infiltrating-lymphocytes-derived atypical TCRs recognizing breast cancer in a MR1-dependent manner”, Hayee et al have reported new data on unconventional T cell receptors (TCR) that recognize tumour cells in a non-HLA restricted manner. This is a continuation of work previously reported in the European Journal of Immunology in 2021 by Yamaguchi et al, where authors used a very efficient technical approach for identifying novel TCRs using recombination and expression in a mouse T cell reporter line (which they call c-FIT). In this previous work, authors identified a number of non-HLA restricted TCRs, which induced T cell responses against MCF7 breast cancer cell line, and seemed to be dependent on MR1 expression. In this paper, authors further characterize the response of these novel TCRs using MR1-associated ligands, and various genetic manipulations of MCF7 HLA or MR1 expression. Authors find that their atypical 10-59 TCR does not bind to MR1-tetramers, responds to breast cancer cells (but not other cancer types), is dependent on MR1 expression for response, not affected by MR1 ligands, and not affected by K43 mutation of MR1. These results seem somewhat contradictory, and thus should be better explained/summarized in the discussions section. Overall, this work builds nicely on the previous reports about these interesting TCRs, though there are some experimental issues that should be addressed and some editing is required to better contextualize the meaning of these results. Authors are commended on assembling interesting scientific data and a well written report; I look forward to reviewing a revised version of the manuscript soon.
Comment 1: The title gives the impression that new TCRs are identified here. Perhaps this could be changed to reflect that previously identified TCRs are being characterized in this paper
|
Response 1: Thank you very much for valuable suggestion. According to the reviewer’s suggestion, we have changed the title to “Characterization of tumor-infiltrating-lymphocytes-derived atypical TCRs recognizing breast cancer in a MR1-dependent manner.”
|
Comment 2: The abstract is perhaps too specific to those with knowledge of MR1 restricted TCRs. Perhaps better explaining the relevance of AC-6-FP and K43 findings would be helpful to make the impact of the work here more readily apparent.
Response 2: We are thankful to reviewer for suggestion that improved our abstract.
According to the reviewer’s suggestion, we rewrite the abstract as following: MHC class I-related 1 (MR1) molecule is a non-polymorphic antigen-presenting molecule that presents several metabolites to MR1-restricted T cells including mucosal-associated invariant T (MAIT) cells. MR1-ligand bind to MR1 molecule by forming a Schiff base with K43 residue of MR1, which induces the folding of MR1 and its reach to the cell surface. An antagonistic MR1-ligand, Ac-6-FP, and K43A mutation of MR1 is known to inhibit the responses of MR1-restricted T cells. In this study, we analyzed MR1-restricted TCRs obtained from tumor-infiltrating lymphocytes (TILs) from breast cancer patients. One of these TCRs (TCR10-59) responded to breast cancer cell lines in an independent manner of microbial infection and did not respond to other cancer cell lines or normal breast cells. Interestingly, the reactivity of the TCR10-59 was not inhibited by Ac-6-FP, while it was attenuated by K43A mutation of MR1. Our findings suggest the existence of a novel class of MR1-restricted TCRs whose antigen is expressed in some breast cancer cells and binds to MR1 dependently on K43 residue of MR1 but without being influenced by Ac-6-FP. This work provides new insight into the physiological role of MR1 and MR1-restricted T cells.
Comment 3: Given that TCR 10-59 expressing cells seem to be dependent on MR1 expression (as per knockout experiments), but do not bind MR1 tetramers, and do not even react to MR1 modulation via antigen treatment, is a possible explanation that TCR 10-59 could react to a MR1-associated receptor??
Response 3: The reviewer suggests the possibility that TCR10-59 reacts to unknown MR1-associated receptor that is expressed on MCF7 cells. We agree that this is one of the possible mechanisms of TCR10-59 response to MCF7 cells. However, according to the reviewer’s comment (9), we analyzed the response of TCR10-59 to Ac-6-FP-treated MCF7 cells. On Ac-6-FP-treated MCF7 cells, MR1-expression was increased, but the response of TCR10-59-expressing BW cells to MCF7 cells was not changed in the presence or absence of Ac-6-FP. If TCR10-59 reacts to MR1-associated receptor, the response of TCR10-59-expressing BW cells to Ac-6-FP-treated MCF7 cells should be augmented, but it did not. This result did not support the reviewer’s hypothesis. But because we have not identified the antigenic molecule of TCR10-59, we still should take the reviewer’s hypothesis into consideration.
Comment 4: Line 249: please provide the specific reference here
Response 4: According to the reviewer’s indication, we added the reference “Eur J Immunol, 51: 2306–2316, 2021”. In revised manuscript, it is in line 255.
Comment 5: Line 270: please clarify how results specifically correlate with the work of Eckle
Response 5: Eckle et al. showed that MR1 expression level on C1R cells was dose-dependently upregulated by incubating the C1R cells with Ac-6-FP and less effectively with 5-OP-RU. Our results were corresponding with these results. It is mentioned in lines 270-271 in revised manuscript.
Comment 6: Line 272: Confirmation of deletion should be described before results with delta-MR1 experiments
Response 6: We have deleted the description of confirmation of MR1-deletion from the results and moved it to material and method section lines 114-115.
Comment 7: Section 3.2: The results discussed here are entirely supplemental. A summary table or figure of this data should be included as a main figure.
Response 7: According to the reviewer’s indication, we added summary tables of MR1, HLA-E and CD1s-expression level in each cell line in Table 2, 3, and 4 respectively in revised manuscript.
Comment 8: Line 265: It appears that with or without IFNg treatment, no expression of MR1 is detected on MCF7 cells, but this is not mentioned in the manuscript. Why is this? Only with AC-6-FP does MR1 expression become apparent on MCF7 cells, and also on other breast cancer cells (S3D), and non-breast cancer cells (S3E)
Response 8: According to the reviewer’s suggestion, we added the description that IFN-γ-treatment did not increase the expression of MR1 on MCF7 cells as shown in Figure S2D. It is mentioned in line 269-270 in revised manuscript.
Comment 9: Given that authors show that in almost all cancer cell lines treatment with Ac-6-FP increases surface MR1 expression, it would be important to test the response of 10-59 TCR expressing cells +/- Ac-6-FP instead of only under IFN-g stimulation. Furthermore, all of the breast cancer cells reported in S3D should tested +/- Ac-6-FP for response to get a better idea of how universal the response on 10-59 is against breast cancer cells. This could be done with either hPBMC or BW clones.
Response 9: We have examined the response of TCR10-59-expressing BW cells to Ac-6-FP-treated MCF7 cells and already added the results as Figure 5B. The response of TCR10-59-expressing BW cells to MCF7 cells was not changed in the presence or absence of Ac-6-FP. According to the reviewer’s suggestion, we also treated the other breast cancer cells with Ac-6-FP and analyzed the response of TCR10-59-expressing BW cells to Ac-6-FP-treated breast cancer cells. As shown in Figure S15, the response of TCR10-59 to breast cancer cells was not affected with the addition of Ac-6-FP. We have mentioned the result in revised manuscript in lines 368-370.
Comment 10: Line 282: The experiment wherein overexpression of patient HLA, HLA-B*59:01 in MCF7 is used should be explained more clearly.
Response 10: In Fig. 2, we used patient’s HLA-dependent TCR2-15 as a representative of HLA-dependent TCRs to show that the other TCRs responses were independent of patient’s HLA. We used HLA-B*59:01 overexpressing MCF7 cells as target cells. We added this explanation in the revised manuscript in lines 287-289.
Comment 11: Line 289: What is a “donor B”
Response 11: In Fig. 2, we expressed TCRs on human PBMCs obtained from two different donors to exclude the possibility of alloreactivity of PBMCs. “Donor B” indicated the second donor. We substituted “donor B” with “another donor”. In revised manuscript, it is mentioned in line 295.
Comment 12: Line 290: A better summary of the specific reasoning for focus on TCR10-59 would be helpful (eg, good expression in PBMC and its strong response characteristics)
Response 12: The response of HLA-independent TCRs (2-78, 2-81, 10-59 and 10-69) expressed on mouse T cells was weaker than the response of TCR10-59 expressed on human PBMCs. For analyzing the cytotoxicity to MCF7 cells, we should use TCR10-59-expressing human PBMCs. Thus, we focused on TCR10-59 for further analysis in detail. In revised manuscript, it is mentioned in lines 296-299.
Comment 13: Line 360: The rationale/background on MR1-K43A should be better introduced to make clear why these experiments were done.
Response 13: According to the suggestion of the reviewer, we added the background on MR1-K43A in lines 378-383.
Comment 14: Section 3.8: This could be integrated into section 3.5
Response 14: According to the reviewer’s suggestion, we integrated section 3.8 into section 3.5.
In revised manuscript, it is in lines 348-353.
Comment 15: Line 385-386: This line does not seem to be needed
Response 15: As suggested by the reviewer, lines 385-386 may be accidentally included. We removed these lines.
Comment 16: In several figures: The control is labeled APC-ve, what does this mean?
Response 16: We abbreviated “negative” as “-ve”. “APC-ve” means TCR-expressing BW cells or T cells were cultured in the absence of antigen presenting cells such as MCF7 cells.
Comment 17: Figure 5: In some subfigures the dose is written nM/ml. Is this nmol/mL or nM?
Response 17: Thank you for the indication. “nM/ml” should be corrected to “nM”. We have corrected them.
Comment 18: Line 641: While one possible explanation seems to be that MR1 may present a breast cancer antigen, the results are not necessarily suggestive of this. Please reword this sentence.
Response 18: According to the reviewer’s suggestion, we reword the sentence of line 641 as follows: One possible explanation of our results is that MR1 can present an antigen produced in some breast cancer cells. In revised manuscript, it is in lines 677-678.
Comment 19: The conclusion section should be written more generally, and provide some more context for how these results might be applied. A future direction to try to identify the specific antigen for 10-59 is a good idea, but how will this be done?
Response 19: According to the reviewer’s suggestion, we rewrite the conclusion section as following: In conclusion, we have reanalyzed the MR1-restricted TCRs that reacted to breast cancer cells and shown that they responded to some breast cancer cell lines but not to other cancer cell lines. The identification of the antigen for our MR1-restricted TCRs will expand our understanding of the physiological role of MR1 and MR1-restricted T cells. To this end, we plan to perform 1) genome-wide knocked out experiment using CRISPR/Cas9 system (Nat Immunol, doi: 10.1038/s41590-019-0578-8. and Sci Immunol, doi: 10.1126/sciimmunol.adn0126) and 2) cDNA library experiment, to identify the molecules that are involved in the production of antigenic molecule or its presentation to MR1 molecules. In revised manuscript, it is in lines 713-719.

Round 2
Reviewer 2 Report
Comments and Suggestions for Authors
Authors have adequately addressed my concerns in their revised manuscript.